# Cost-effective complete genome sequencing using the MinION platform for identification of recombinant enteroviruses

Yeh-Sheng Chien,[1] Feng-Jui Chen,[1,2] Han-Chieh Wu,[1] Chieh-Hua Lin,[3] Wen-Chiung Chang,[1] David Perera,[4] Jyh-Yuan Yang,[5] Min-Shi Lee,[1] Yu-Chieh Liao[3]

**ABSTRACT** Enteroviruses (EVs) are a group of viruses that cause various human illnesses. While the CODEHOP (COnsensus-DEgenerate Hybrid Oligonucleotide Primer) method can generate VP1 gene fragments for enterovirus genotyping, it is unable to detect recombinant strains. Recent advances in viral genome sequencing using next-generation sequencing technologies have enabled comprehensive analyses. However, the high cost poses a challenge for widespread adoption. To address this issue, this study proposes a cost-effective approach for generating complete enterovirus genome sequences using the Oxford Nanopore MinION sequencer. This protocol not only facilitates the generation of accurate genome sequences for various enterovirus strains but also allows for the differentiation of co-infections from viral isolates. In addition, the method can generate polyprotein sequences as well as peptide sequences of the upstream ORF (uORF) whose expression can impact virus infection. Through the analysis of complete enterovirus genomes, this study successfully identified seven enterovirus A71 isolates obtained during the 2018 enterovirus outbreak in Malaysia and Taiwan as recombinants between enterovirus A71 and coxsackievirus A2. Furthermore, our study has made a significant discovery by establishing a strong correlation between uORF trees and the epidemics of EVA71. This finding highlights the potential of uORF sequences as valuable indicators for monitoring and understanding the spread of EVA71 infections. We also identified notable amino acid changes in the transmembrane domain of the uORF protein within a newly identified lineage. These findings provide crucial insights into the molecular characteristics and evolutionary dynamics of EVA71, offering valuable information for future research and intervention strategies.

**IMPORTANCE** By employing a cost-effective approach for complete genome sequencing, the study has enabled the identification of novel enterovirus strains and shed light on the genetic exchange events during outbreaks. The success rate of genome sequencing and the scalability of the protocol demonstrate its practical utility for routine enterovirus surveillance. Moreover, the study's findings of recombinant strains of EVA71 and CVA2 contributing to epidemics in Malaysia and Taiwan emphasize the need for accurate detection and characterization of enteroviruses. The investigation of the whole genome and upstream ORF sequences has provided insights into the evolution and spread of enterovirus subgenogroups. These findings have important implications for the prevention, control, and surveillance of enteroviruses, ultimately contributing to the understanding and management of enterovirus-related illnesses.

**KEYWORDS** enterovirus, recombinant, next-generation sequencing, nanopore, upstream ORF

Enteroviruses (EVs) belong to the *Picornaviridae* family and are widespread and environmentally stable RNA viruses (1). They are characterized by a conserved

Address correspondence to Yu-Chieh Liao, jade@nhri.edu.tw, or Min-Shi Lee, minshi.lee@hotmail.com.

Yeh-Sheng Chien and Feng-Jui Chen contributed equally to this article. The author order was determined based on seniority.

The authors declare no conflict of interest.

See the funding table on p. 13.

genomic structure, consisting of a single-stranded RNA genome ranging in size from 7.2 to 8.5 kb (2). The capsid proteins, VP1 to VP4, are encoded in the P1 region of the enterovirus genome, whereas the nonstructural proteins are encoded in the remaining P2 and P3 regions (3). The VP1 capsid gene sequencing has shown a strong correlation with serotypes determined through virus neutralization tests, making it an excellent tool for genotyping enteroviruses. To facilitate the sequencing of VP1 sequences, a consensus degenerate hybrid oligonucleotide primer (CODEHOP) approach was proposed (4, 5). In 2019, the existence of an upstream open reading frame (uORF) that encodes an additional polypeptide called the uORF protein (UP) was proposed and demonstrated to play a role in modulating gut infection (1). However, research on UP is very limited, and its evolution remains unclear. The *Enterovirus* genus encompasses 15 species, namely *Enterovirus A–D* (human enteroviruses), *Enterovirus E–L* and *Rhinovirus A–C*, as defined by the International Committee on Taxonomy of Viruses (ICTV) (6). These species contain over 200 serotypes, which are associated with a range of diseases. For example, Coxsackievirus A16 and Enterovirus A71 are known to cause hand-foot-mouth disease (7), while Polioviruses 1–3 are responsible for poliomyelitis (6). In addition, enterovirus D68 and various rhinoviruses are linked to pneumoniae (8).

RNA viruses, including enteroviruses, are characterized by high mutation rates, leading to the generation of new genetic variants. In addition, these viruses can undergo genome recombination, a significant driving force in their evolution. Recombination events can result in the emergence of novel enterovirus variants with increased pathogenicity and fitness, contributing to the dynamic nature of these viruses (9, 10). In particular, intertypic recombination plays a crucial role in the emergence of highly pathogenic circulating vaccine-derived polioviruses. These recombinant viruses have been responsible for numerous outbreaks of paralytic poliomyelitis worldwide (11). Enteroviruses have been found to exhibit species-specific differences (12), with *Enterovirus B* being the most detected species globally, *Enterovirus A* being a more common species in Asia, and *Enterovirus C* being more prevalent in Africa. Although outbreaks typically display serotype-specific dominance, it is important to note that the co-circulation of multiple serotypes can occur worldwide (13–16). To gain a comprehensive understanding of enteroviruses, it is crucial to obtain complete enterovirus genome sequences. These genome sequences provide valuable insights into the diversity, epidemiology, evolution, and pathogenicity of enteroviruses. However, achieving complete genome sequencing can be challenging due to their high variability and dynamic nature in epidemics. Challenges arise in primer design to ensure coverage of diverse strains and in managing the sequencing costs associated with large-scale genomic studies (2, 17, 18).

In recent years, the MinION sequencer from Oxford Nanopore Technologies (ONT) has gained widespread popularity for viral genome sequencing (18–24). This is primarily attributed to its portability and cost-effectiveness compared to other sequencing technologies. In our previous study (2), we employed Illumina sequencing technology to sequence the genome of 52 enterovirus isolates. However, the cost associated with constructing genome libraries using this method was not sufficiently low to make it suitable for widespread use in virus surveillance. In this study, we aimed to address this limitation by providing a cost-effective protocol for generating complete enterovirus genomes using a MinION sequencer. Our protocol also enables the generation of polyprotein sequences and peptide sequences of the uORF. Through comprehensive phylogenetic analyses of the whole genome and uORF, we investigated the epidemics of enterovirus A71 subgenogroups C1 and B5 circulating in Taiwan in 2019 and in Malaysia in 2018.

## RESULTS

### The pilot run of MinION sequencing

In this study, a barcoding PCR protocol was proposed to amplify the full-length enterovirus genome. The amplification was achieved using pan-EV forward primers and ONT universal tags as reverse primers (Fig. 1). After native barcoding and adaptor ligation, the prepared library with samples in dual barcodes was sequenced using MinION. During the pilot run, the MinION sequencer generated 7,662,562 reads (around 16 Gbp). However, more than half of these reads (4,146,978) were shorter than 1,000 bp. To ensure the quality of the reads, a filtering process was applied to retain reads with a length between 1,000 and 8,000 bp, resulting in 13.4 Gbp (83.75%) for further analysis. As shown in Table 1, 36 enterovirus isolates were sequenced, including 26 isolates from species A, 9 from species B, and 1 isolate from species C. Of these, 18 isolates (P01–P18) had previously been sequenced using the Illumina platform (accession numbers starting with MF, in Table 1) (2). The comparison between the genome sequences obtained from MinION and Illumina platforms revealed exceptionally high average nucleotide identities of 99.98%, as indicated in Table 1.

Among the 18 new samples (P19–P36), 17 were successfully sequenced using the MinION platform, with 14 samples showing high nucleotide identity (96%–100%) to virus sequences in the GenBank database. However, three samples (P22, P23, and P35) had lower nucleotide identity (90%–94%), warranting further analysis for potential recombination events. The genome sequence of one virus isolate (P33) was not obtained due to the generation of only 39 filtered reads (Table S1). This sample had a high Ct of 38 in initial qRT-PCR tests, which may have contributed to the limited number of reads obtained. Our proposed pipeline incorporates the filtering process that enabled us to generate 35 complete viral genomes in this pilot run. However, the significant

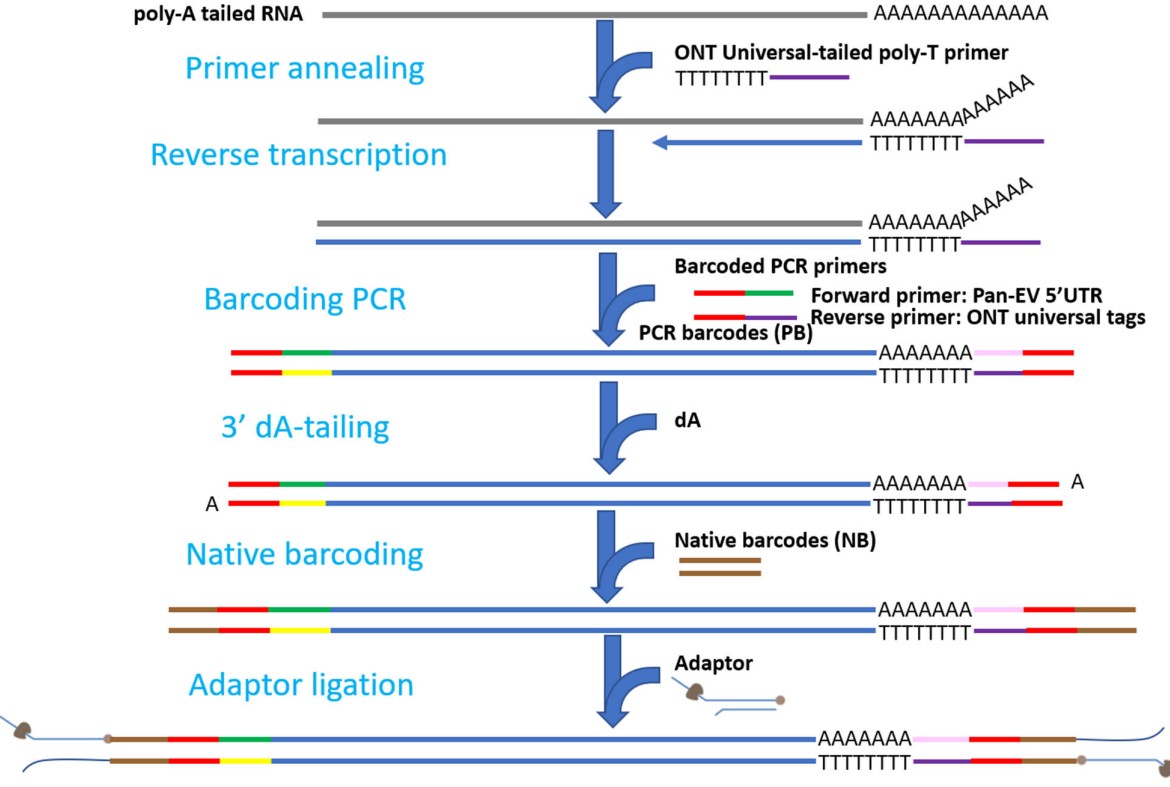

**FIG 1** A schematic workflow for enterovirus whole-genome sequencing. Green and purple lines represent pan-enterovirus 5'UTR forward primer sequence— CCCTGAATGCGGCTAATCCTAA—and ONT universal reverse tag sequence—ACTTGCCTGTCGCTCTATCTTC, respectively. Red and brown lines represent barcode sequences for PCR and native barcoding.

**TABLE 1** MinION results of a pilot run for genome sequencing of enteroviruses

| ID | Sp[a] | Ct | MinION | Illumina[b] | Accession (Blast hit) | Identity (%) | Country | Year |
|---|---|---|---|---|---|---|---|---|
| P01 | A | 23 | CVA2 | CVA2 | MF422535 | 99.97 | Taiwan | 2008 |
| P02 | A | 24 | CVA2 | CVB4/**CVA2** | MF422542 | 99.99 | Taiwan | 2008 |
| P03 | A | 19 | CVA2 | **CVA2**/E30 | MF422543 | 99.97 | Taiwan | 2008 |
| P04 | A | 26 | CVA4 | CVA4 | MF422544 | 99.97 | Taiwan | 2008 |
| P05 | A | 30 | CVA5 | CVA5 | MF422547 | 99.99 | Taiwan | 2008 |
| P06 | A | 19 | CVA5 | **CVA5**/E3 | MF422548 | 99.99 | Taiwan | 2008 |
| P07 | A | 22 | CVA6 | **CVA6**/E30 | MF422553 | 99.91 | Taiwan | 2008 |
| P08 | A | 24 | CVA6 | CVA6 | MF422554 | 100 | Taiwan | 2008 |
| P09 | A | 26 | CVA10 | CVA10 | MF422532 | 99.99 | Taiwan | 2008 |
| P10 | A | 28 | CVA16 | CVA16 | MF422533 | 99.98 | Taiwan | 2008 |
| P11 | B | 30 | CVA9 | CVA9 | MF422557 | 100 | Taiwan | 2008 |
| P12 | B | 18 | E3 | CVB4/**E3** | MF422572 | 99.99 | Taiwan | 2008 |
| P13 | B | 27 | E3 | E3 | MF422570 | 99.99 | Taiwan | 2008 |
| P14 | B | 17 | E3 | CVA5/**E3** | MF422568 | 99.97 | Taiwan | 2008 |
| P15 | B | 29 | E6 | E6 | MF422579 | 99.97 | Taiwan | 2008 |
| P16 | B | 30 | E9 | E9 | MF422580 | 100 | Taiwan | 2008 |
| P17 | B | 15 | E30 | CVA6/**E30** | MF422573 | 99.97 | Taiwan | 2008 |
| P18 | B | 14 | E30 | CVA2/**E30** | MF422577 | 99.94 | Taiwan | 2008 |
| P19 | A | 21 | EVA71/B5 | - | (DQ341363) | 99.68 | Malaysia | 2003 |
| P20 | A | 22 | EVA71/B5 | - | (DQ341363) | 99.81 | Malaysia | 2003 |
| P21 | A | 24 | EVA71/B5 | - | (DQ341363) | 99.77 | Malaysia | 2003 |
| P22 | A | 23 | EVA71/B5 | - | (OM417111) | 93.73 | Malaysia | 2018 |
| P23 | A | 19 | EVA71/B5 | - | (OM417111) | 93.67 | Malaysia | 2018 |
| P24 | A | 22 | EVA71/B5 | - | (KF974790) | 96.55 | Malaysia | 2018 |
| P25 | A | 22 | EVA71/B5 | - | (LC627079) | 99.4 | Taiwan | 2016 |
| P26 | A | 15 | EVA71/B5 | - | (MG674288) | 100 | Taiwan | 2019 |
| P27 | A | 30 | EVA71/B5 | - | (DQ341363) | 97.45 | Vietnam | 2013 |
| P28 | A | 32 | EVA71/B5 | - | (MH716268) | 99.74 | Vietnam | 2015 |
| P29 | A | 30 | EVA71/B5 | - | (MH716268) | 99.52 | Vietnam | 2015 |
| P30 | A | 18 | EVA71/B5 | - | (LC627081) | 99.31 | Vietnam | 2017 |
| P31 | A | 23 | EVA71/C1 | - | (DQ341358) | 97.84 | Malaysia | 2003 |
| P32 | A | 20 | EVA71/C1 | - | (DQ341358) | 97.81 | Malaysia | 2003 |
| P33 | A | 38 | EVA71/C4 | - | - | | Taiwan | 2015 |
| P34 | A | 30 | EVA71/C4 | - | (LC627087) | 98.52 | Vietnam | 2016 |
| P35 | B | 30 | E11 | - | (KY981558) | 90.95 | Taiwan | 2003 |
| P36 | C | 26 | CVA24 | - | (KR478685) | 98.54 | Vietnam | 2013 |

[a]Enterovirus species.
[b]Genome has been sequenced using the Illumina platform in the previous study, and one of the co-infected species was identified and hightlighted in bold in this study.

presence of short reads suggests that implementing a size-selection procedure during the amplification and construction of the nucleotide library could further improve the quality of the generated sequences.

## Highly accurate enterovirus genome sequence generation with MinION

The SPRI size selection technique was employed to avoid unnecessary sequencing of short amplicon products. After completing the size-selection process, the sequencing proportion of long reads increased considerably from 45.88% to 68.78%. A total of 70 enterovirus isolates were sequenced in this high-throughput run (Table S2), including 33 samples (H01–H13 and H22–H41) that were previously sequenced using the Illumina platform (2), and 8 samples (H14–H21) that were sequenced in the pilot run. The average nucleotide identity remained consistently high at 99.98%, both in comparing the corresponding genome sequences obtained between the Illumina and MinION platforms and between the pilot and the high-throughput runs. The 59 re-sequencing

**TABLE 2**  Re-sequencing samples covering 16 genotypes

| Species | Genotype | No. | Sample ID |
|---------|----------|-----|-----------|
| A | Coxsackievirus A2, CVA2 | 6 | <u>H01</u>[a]–H03, P01–P03 |
|   | Coxsackievirus A4, CVA4 | 5 | <u>H04</u>–H07, P04 |
|   | Coxsackievirus A5, CVA5 | 3 | <u>H08</u>, P05–P06 |
|   | Coxsackievirus A6, CVA6 | 5 | <u>H09</u>–H11, P07–P08 |
|   | Coxsackievirus A10, CVA10 | 3 | H12–<u>H13</u>, P09 |
|   | Coxsackievirus A16, CVA16 | 1 | P10 |
|   | Enterovirus A71, EVA71 | 8 | H14-H21[b] |
| B | Coxsackievirus A9, CVA9 | 3 | H22–<u>H23</u>, P11 |
|   | Coxsackievirus B4, CVB4 | 3 | <u>H24</u>–H26 |
|   | Echovirus E3, E3 | 7 | <u>H27</u>–H30, P12–P14 |
|   | Echovirus E6, E6 | 2 | <u>H31</u>, P15 |
|   | Echovirus E9, E9 | 3 | <u>H32</u>–H33, P16 |
|   | Echovirus E25, E25 | 3 | <u>H34</u>–H36 |
|   | Echovirus E30, E30 | 5 | <u>H37</u>–H39, P17–P18 |
| C | Poliovirus 1, PV1 | 1 | <u>H40</u> |
| D | Enterovirus D68, EVD68 | 1 | <u>H41</u> |

[a]Samples used for the simulation of the mixture are underlined.
[b]These eight samples were derived from aliquots of the samples used in the pilot run (P19–24, P31–32).

samples in total (P01–P18 and H01–H41) contain 4 human enterovirus species A–D and 16 serotypes, as shown in Table 2. This indicates that the sequencing protocol can effectively cover the diverse enterovirus strains, and the resulting genome sequences are accurate. In addition to the 41 re-sequencing samples, 27 out of the remaining 29 were successfully sequenced using the MinION platform, resulting in 29 viral genome sequences (Table 3). It is important to note that complete genomes were not obtained for two samples, H43 and H64, which may be attributed to their extremely low cDNA concentration of 0.188 and 1.54 ng/mL, respectively. Besides, multiple genomes were produced from the two co-infection samples (H50 and H57). As listed in Table 3, among the 29 viral genomes, 24 exhibited high nucleotide identity (96%–100%) with sequences from the GenBank database, but 5 genomes (H54, H57-2, H59, H62, and H63) showed lower nucleotide identity (93%–95%). It should be noted that while the two genome sequences for the H57 sample were automatically generated using the default settings in the nanoEV pipeline, the H50 sample required an increase in the number of long reads from 200 to 500 to differentiate multiple genomes. Overall, the proposed protocol allows us to obtain accurate genome sequences for various enterovirus strains using the MinION platform.

## Co-infection detection from a simulation run

The production of multiple genomes from a single sample indicates that our pipeline can identify co-infections. To further evaluate this capability, we conducted a simulation of co-infection by merging sequencing reads from two distinct sample types. In all, 16 samples were chosen for each serotype, as indicated by underline in Table 2 and Table 3, to generate 120 co-infection combinations. We analyzed the 120 co-infection combinations using the nanoEV pipeline, resulting in 87 single-genome and 33 double-genome identifications (Fig. 2). Although all reads from both samples were combined for the mixture simulation, only the longest 200 reads were selected for analysis in the nanoEV pipeline. The uneven proportions of reads from each sample are represented by the dark red and blue colors in Fig. 2. As expected, the double-genome was mainly identified in equal proportions, as depicted by the white cells. Notably, reads from H09 (CVA6) were dominant in the mixtures, as indicated by the red row, suggesting that this sample has longer reads than the others. This suggests that H09 had longer reads than the other samples, resulting in consistent identification of a CVA6 genome from the mixture containing H09. By contrast, H23 (CVA9) reads were less prevalent in the

**TABLE 3** MinION results for genome sequencing of new enterovirus samples

| ID | Qbit | MinION | Accession | Identity (%) | Country | Year |
|---|---|---|---|---|---|---|
| H42[a] | 1.51 | CVA16 | OP562200 | 98.88 | Taiwan | 2018 |
| H43 | 0.188 | CVA16 | | | Vietnam | 2019 |
| H44 | 1.95 | CVA4 | MF422544 | 99.97 | Taiwan | 2008 |
| H45 | 13.3 | CVA5 | MF422549 | 99.99 | Taiwan | 2008 |
| H46 | 2.86 | CVA5 | MF422549 | 99.93 | Taiwan | 2008 |
| H47 | 236 | EVA71/B4 | AF352027 | 99.3 | Malaysia | 2000 |
| H48 | 29.2 | EVA71/B4 | DQ341366 | 98.95 | Malaysia | 2000 |
| H49 | 253 | EVA71/B4 | AF316321 | 99.17 | Malaysia | 2000 |
| **H50-1**[b] | 200 | EVA71/B5 | DQ341363 | 99.35 | Malaysia | 2003 |
| **H50-2** | | EVA71/B4 | AF352027 | 99.25 | | |
| H51 | 300 | EVA71/B5 | DQ341363 | 99.77 | Malaysia | 2003 |
| H52 | 408 | EVA71/B5 | DQ341363 | 99.7 | Malaysia | 2003 |
| H53 | 106 | EVA71/B5 | DQ341362 | 99.93 | Malaysia | 2003 |
| H54 | 156.4 | EVA71/B5 | OM417111 | 93.58 | Malaysia | 2018 |
| H55 | 200 | EVA71/B5 | KF974790 | 96.51 | Malaysia | 2018 |
| H56 | 459.6 | EVA71/B5 | KF974790 | 96.53 | Malaysia | 2018 |
| **H57-1** | 30.4 | EVA71/B5 | KF974790 | 96.53 | Malaysia | 2018 |
| **H57-2** | | EVA71/B5 | MG756694 | 93.96 | | |
| H58 | 166.8 | EVA71/B5 | KF974790 | 96.55 | Malaysia | 2018 |
| H59 | 480 | EVA71/B5 | OM417111 | 93.71 | Malaysia | 2018 |
| H60 | 312 | EVA71/B5 | KF154354 | 96.57 | Malaysia | 2018 |
| H61 | 54 | EVA71/B5 | KF974790 | 96.48 | Malaysia | 2018 |
| H62 | 440 | EVA71/B5 | OM417111 | 93.72 | Malaysia | 2018 |
| H63 | 89.2 | EVA71/B5 | MG756694 | 94.62 | Taiwan | 2018 |
| H64 | 1.54 | EVA71/B5 | | | Vietnam | 2019 |
| H65 | 30.8 | EVA71/C1 | KX139462 | 97.37 | Taiwan | 2019 |
| H66 | 20 | EVA71/C1 | KX139462 | 97.27 | Taiwan | 2019 |
| H67 | 68 | EVA71/C4 | KU936128 | 98.59 | Taiwan | 2015 |
| H68 | 286 | E3 | MF422572 | 99.99 | Taiwan | 2008 |
| H69 | 236 | E9 | MF422580 | 99.94 | Taiwan | 2008 |
| H70 | 216 | E30 | MF422578 | 99.98 | Taiwan | 2008 |

[a]These two underlined samples were used for the simulation of the mixture.
[b]Multiple genomes were identified in the bold samples.

mixtures, as indicated by the blue row, making it easier to detect the other genome when mixed with H23. In comparing the 153 generated sequences (Table S3) to the corresponding reference sequences and the consensus sequences of H42 and H54, we found that the average sequence identity ranged from 99.92% (for enterovirus A71) to 100% (for Coxsackievirus A6 and Echovirus E3). Despite observing a maximum sequence error of 0.26% in the genome generation of E9 (as mixed with E3), we still obtained an average sequence identity of 99.98% for the 153 sequences generated from the mixtures.

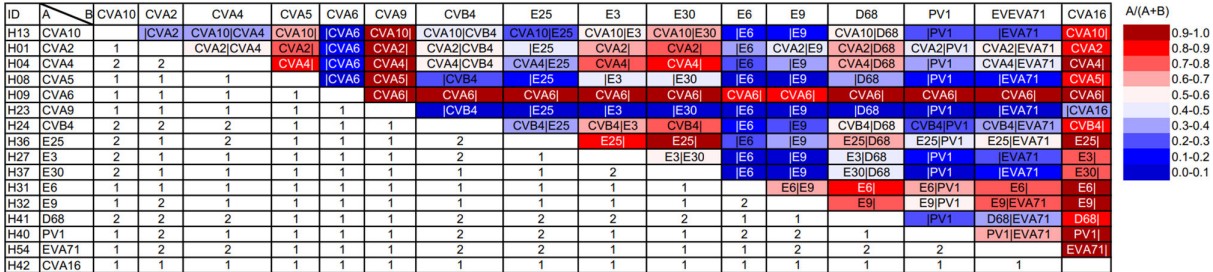

**FIG 2** Single- and double-genome identification of a mixture simulation. The uneven proportions of reads are represented by red and blue colors. The upper-right triangle indicates the identified types, and the lower-left triangle indicates the number of identified genomes.

These findings demonstrate that our proposed protocol, in conjunction with the nanoEV pipeline, can generate highly accurate enterovirus genomes even from viral mixtures.

## Correlation between epidemiology and phylogenetic trees of EVA71

EVA71 is a major cause of hand-foot-mouth disease in children and is known to be associated with severe neurological complications. Its circulation exhibits notable dynamics in the Asia-Pacific region (7). According to reports from references (25, 26), subgenogroups B5 and C1 of EVA71 emerged in Thailand in 2017 and Taiwan in 2018, respectively, but these two studies did not conduct genomic sequencing to clarify the evolution of these novel viruses. Our study included two 2019 C1 genomes from Taiwan (H65 and H66 in Table 3). Notably, the phylogenetic tree based on VP1 (Fig. 3A) revealed the close phylogenetic relationship between the 2018 (25) and the 2019 C1 viruses isolated in Taiwan. Moreover, the phylogenetic tree based on the whole genome obtained in this study (Fig. 3B) demonstrates that viruses isolated after 2014 form a distinct lineage from those reported during 1991–2010. While this observation was previously documented in reference (27), this study is the first to present a similar phylogenetic analysis based solely on the nucleotide sequences of the uORF (Fig. 3C). The UP, recognized for its role in modulating virus infection in gut epithelia (1), was employed to compare the peptide sequences of these two lineages (Fig. 3D). As illustrated in Fig. 3D, the insertion of 11 amino acids in the novel C1 viruses isolated after 2014 may confer an advantage for viral spread. Therefore, in addition to the fact that the upsurge of neurologic disease in France in 2016 was associated with enterovirus infection with the multirecombinant C1 lineage (27), our study provides the unique UP sequences consisting of 75 amino acids (Fig. 3D) obtained from the new C1 outbreak. Three amino acid mutations (T34S, K36T, and L45V) were identified in the predicted transmembrane domain (site 22–site 45) of UP.

EVA71 subgenogroup B5 has been reported to cause severe infections in Thailand in 2017 (26) and continues to circulate in Vietnam (28). However, the molecular epidemiology of the recent B5 viruses remains unclear. In our study, 27 viral genomes of EVA71/B5 were obtained (P19–P30 in Table 1 and H50-1, H51–H63 in Table 3) from Malaysia, Taiwan, and Vietnam spanning the period from 2003 to 2018. Analyzing the VP1 tree (Fig. 4A), we observed that the 2018 Malaysia isolates form two distinct lineages. The lineage highlighted with a background color exhibited a closer phylogenetic relationship with the majority of isolates from Thailand and Vietnam (26, 28). Despite the limited number of available whole-genome sequences of EVA71 B5 (29), our whole-genome analysis (Fig. 4B) clearly showed that the seven B5 isolates formed a new lineage. The distinct lineages can also be observed in the uORF tree (Fig. 4C). As depicted in Fig. 4D, two amino acid differences, V38I and I39T, were observed in the transmembrane domain of UP. These findings indicate a strong correlation between uORF trees and the epidemics of EVA71, suggesting that the amino acid changes in the transmembrane domain of UP may facilitate virus release. However, further experimental investigations are necessary to validate these observations.

## Enterovirus recombination analysis

In this study, eight isolates (P22, P23, and P35 in Table 1 and H54, H57-2, H59, H62, and H63 in Table 3) were identified with low genome identities (<95%) when compared to virus sequences in the GenBank database. As shown in Fig. 4B, the whole-genome analysis revealed that seven B5 isolates (P22, P23, H54, H57-2, H59, H62, and H63) formed a distinct lineage from other B5 strains, suggesting that they were potential recombinant strains. To investigate their genome structure, similarity plot and bootscan analyses were performed using SimPlot (30). The results (Table S5) indicated a genomic recombination event, where the nucleotide sequences of these isolates were similar to a B5 strain (MG756694, Taiwan 2015) up to nucleotide 5400, but became genetically closer to a CVA2 strain (MH111016, Australia 2016) beyond that point. Our analysis confirmed these

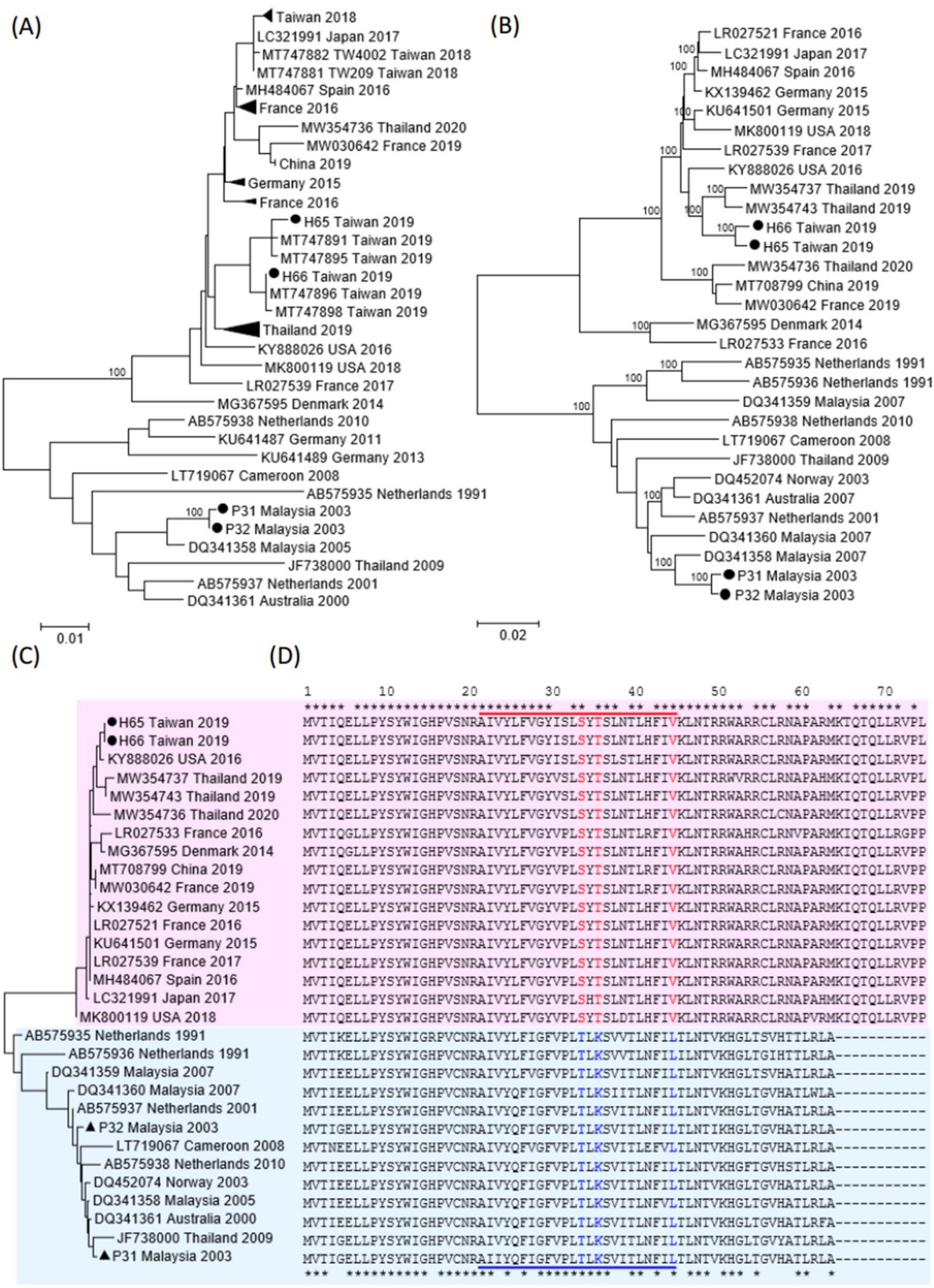

**FIG 3** Phylogenetic analyses of EVA71 C1 strains based on (A) VP1 gene sequences, (B) complete genome sequences, and (C) uORF nucleotide sequences. Black circles represent the isolate sequences obtained in this study. Triangles represent condensed strain accessions: Taiwan 2018 (MT747883–90), France 2016, (LR027521–2 and LR027534–8), Germany 2015 (KU641502–3, KU641508, and KX139462), France 2016, (LR027524, LR027531, and LR027546), and Thailand 2019 (MW354737–46). (D) Consensus protein sequences of uORF for the two lineages. The transmembrane domain region is indicated by underlining.

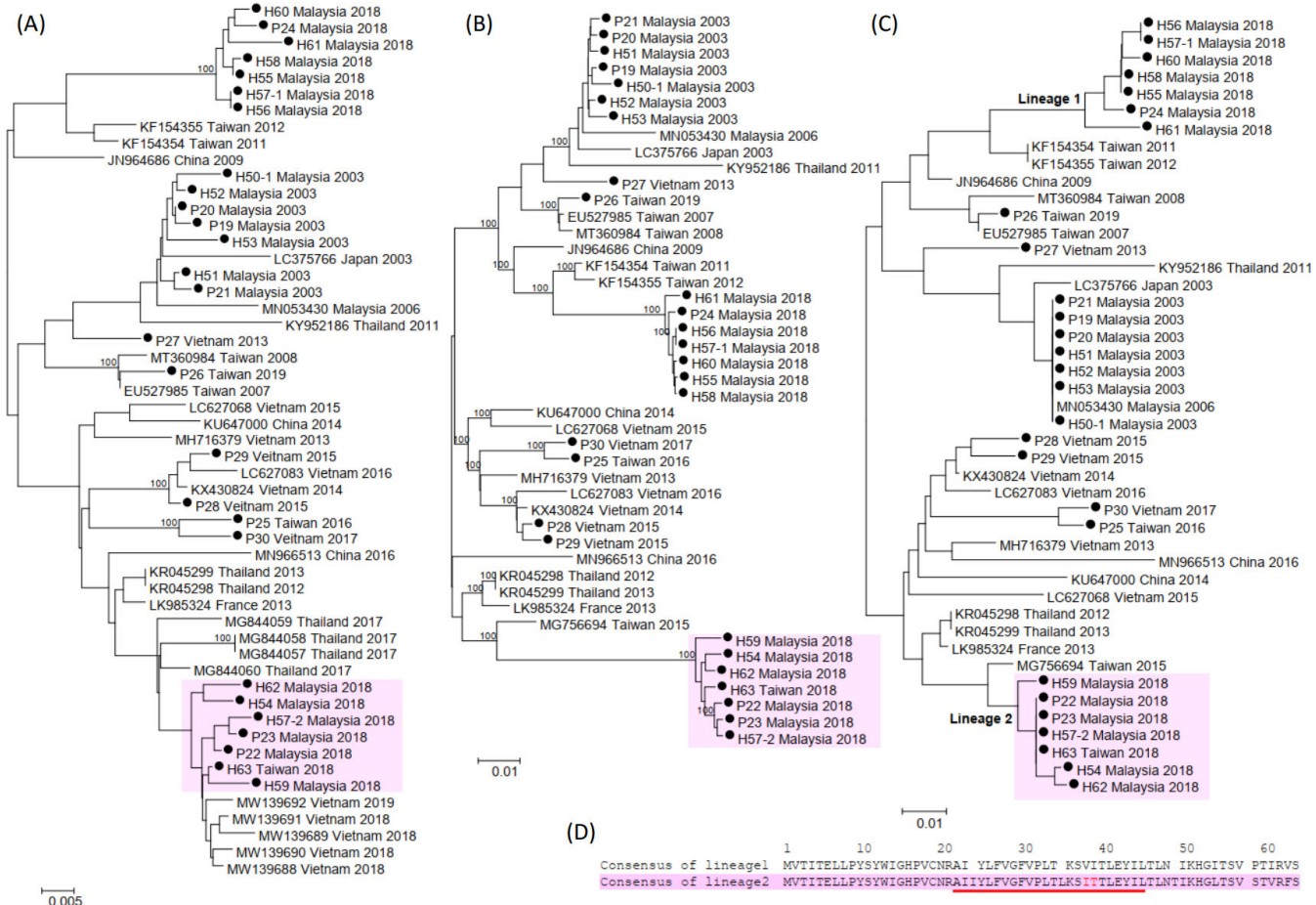

**FIG 4** Phylogenetic analyses of EVA71 B5 strains based on (A) VP1 gene sequences, (B) complete genome sequences, and (C) uORF nucleotide sequences. Black circles represent the isolate sequences obtained in this study. (D) Consensus protein sequences of uORF for the two lineages. The transmembrane domain region is indicated by underlining.

seven EVA71/B5 isolates were recombinants of EVA71 and CVA2, which played a role in the epidemic that occurred in Malaysia and Taiwan in 2018.

## DISCUSSION

Enteroviruses are highly prevalent and contagious viruses that can survive and remain stable in the environment. However, their high variability and epidemic dynamics pose challenges for whole-genome sequencing. Despite utilizing Illumina sequencing technology to sequence over 50 enterovirus isolates in 2019 (2), it should be noted that the Illumina approach proved to be relatively expensive, with a price of around USD500 per sample when outsourcing the sequencing process (2). In this study, we introduced a sequencing protocol that was able to generate complete genome sequences of 16 different enterovirus serotypes included in the study. This protocol was proved to be cost-effective, with a consumption expense of approximately USD30 per sample for all consumables (Table S6), and produced highly accurate genome sequences, achieving a 99.98% accurate rate. In contrast to the random primers employed in the Illumina system, this study used a single pan-enterovirus forward primer that targets the conserved 5′UTR. However, the primer's varying levels of specificity for different types of enteroviruses may need to be considered. For instance, the relatively low cDNA concentrations of H42 and H43 (as shown in Table 3, with Qbit values of 1.51 and 0.188, respectively) could be attributed to the reduced primer efficiency toward CVA16. Although our prior work has documented the co-infection identified in the eight samples

using the Illumina platform (P02, P03, P06, P07, P12, P14, P17, and P18 in Table 1) (2), only a single genome was identified for each sample using the MinION platform. Apart from the possibility of varying primer efficiency, it is also plausible that a single virus became dominant after several passages in cell cultures. Nevertheless, our protocol is capable of producing the dominant virus genomes that are consistent with the virus types identified using the VP1-CODEHOP method.

With the MinION platform along with our nanoEV pipeline, we detected co-infections in two samples (H50 and H57), resulting in four enterovirus A71 sequences: H50-1, H50-2, H57-1, and H57-2 (Table 3). Upon comparing the corresponding pairs, genome sequence identities of 94.11 and 89.63%, polyprotein sequence identities of 99.36 and 98.04%, and uORF sequence identities of 92.19 and 87.50% were obtained for H50 and H57, respectively. The obvious genetic differences suggest the likeness of true co-infection with EVA71, although the genome accuracy could not be confirmed for these four strains. We therefore simulated mixtures by merging reads from two distinct virus types, the 153 generated genomes were found to be highly accurate (99.98%) as they closely matched the expected genome sequences. In an attempt to identify multiple genomes, we increased the default number of long reads from 200 to 500 in the simulation run. However, the outcome (Table S4) was similar to the previous run, as we still produced 152 genomes with a consistent average identity of 99.98%. Considering the increased computational time, we recommend using 200 long reads as the default selection.

We conducted 16 duplicate virus isolates separately, during the pilot and high-throughput runs, and acquired nucleotide with a high degree of consistency (99.97% average identity), along with polyprotein with complete identity (100%). However, two inconsistent uORFs were generated for P11 and P16, corresponding to H23 and H33, respectively. The discontinuous coverage at the beginning of these two representative reads in the pilot run suggests that these reads of chimeric cDNA may have been produced during PCR amplification. While the SPRI size selection technique has been utilized to eliminate short amplicon products in the high-throughput run, it is advisable to prioritize the design of forward primers in future studies.

In this study, we employed the CODEHOP method to identify enterovirus strains and selected 36 enteroviruses representing 14 serotypes for a pilot run of MinION sequencing. The success rate for genome sequencing was 97.2% (35 out of 36 samples). Subsequently, we modified the protocol and successfully scaled up to 70 samples, with only two genomes missed due to the low concentrations of cDNA. By performing a simulation run of co-infection, our protocol was validated to identify double genomes as expected. Utilizing the genome sequences generated in this study, phylogenetic analyses of the whole genome and uORF were performed to investigate the epidemics of enterovirus A71 subgenogroups C1 and B5 circulating in Taiwan in 2019 and Malaysia in 2018, respectively. We thereby identified seven EVA71 B5 isolates as recombinants between EVA71 and CVA2, shedding light on the genetic exchange events during the outbreaks. In addition, our study presented a novel finding by demonstrating a strong correlation between uORF trees and the epidemics of EVA71, along with the identification of amino acid changes in the transmembrane domain of uORF protein in a new lineage. Overall, the enterovirus genome sequences generated using our method can serve as valuable tools for the identification of novel enteroviruses in routine enterovirus surveillance. These findings are critical for enterovirus surveillance and hold potential significance in the prevention and control of enteroviruses.

## MATERIALS AND METHODS

### Clinical virus isolates

As previously described (2), viruses were isolated from various clinical specimens, including throat swabs, nasopharyngeal aspirates, blood, cerebral spinal fluid, and rectal swabs. We first identified virus types using the VP1-CODEHOP method and then selected

samples of interest for genome sequencing using the MinION platform. In the pilot run, we selected 36 samples (14 serotypes) for genome sequencing, including 18 samples that had been previously sequenced using the Illumina method (2) (Table 1). We then modified the process and scaled up to 70 samples, 41 (H01–H41 in Table 2) of which had been previously sequenced using the Illumina method or conducted in the pilot run. In addition, 24 samples from Malaysia, 8 samples from Vietnam, and 15 samples from the Taiwan Centers for Disease Control were sequenced in the pilot run (P19–P36 in Table 1) and the high-throughput run (H42–H70 in Table 3).

## Virus RNA extraction

After observing the cytopathic effect (CPE) in cultured cells, the cells were scraped and centrifuged at $3,000\times g$ for 10 min. To inactivate the virus, the virus supernatant (2 mL) was treated with 0.05% formalin for 8 h. The treated supernatant was then filtered using a 0.22-µm filter to remove cell debris. Viral RNA was extracted using the QIAamp Mini Viral RNA Extraction Kit (Qiagen, Germany).

## CODEHOP method

The enterovirus VP1 gene (350–400 bp) was amplified using a previously described method (31). The amplified DNA was sequenced using the ABI 3730 XL DNA Analyzer (Applied Biosystems, Foster City, CA). Nucleotide sequences of the partial VP1 gene were analyzed using a BLASTN search (32) against the GenBank database for enterovirus typing with the highest identity.

## Reverse transcription

Virus cDNA was generated using SuperScript IV reverse transcriptase (Thermo Fisher Scientific) and a polyT-VN primer (5′-ACTTGCCTGTCGCTCTATCTTC-(dT)20VN). The cDNA synthesis procedure included mixing 1 µL of dNTPs (10 mM each), 2 µL of the polyT-VN primer (2 µM), and 10 µL of extracted RNA, incubating the mixture at 65℃ for 5 min and then snapped cooled on ice for at least 1 min. A mixture of 7 µL of RT buffer containing 4 µL of 5X SSIV buffer, 1 µL of DTT (100 mM), 1 µL of RNaseOUT (40 U/µL), and 1 µL of SuperScript IV (200 U/µL) was added to the pre-cooled RNA mix and incubated at 55℃ for 10 min followed by a 10 min incubation at 85℃. Purified cDNA was recovered using the SPRI size selection method provided on the nanopore community with a final elution volume of 30 µL.

## Barcoding PCR

The full-length genome of each enterovirus sample was PCR amplified and barcoded using the same barcode-labeled forward and reverse primer pair (pan-enterovirus 5′UTR primer: 5′-CCCTGAATGCGGCTAATCCTAA (33) and 5′-ACTTGCCTGTCGCTCTATCTTC, shown in Fig. 1). In all, 12 barcode-labeled primer sets were designed (PB13-PB24). Barcoding PCR was performed by mixing 30 µL of LongAmp Taq 2X Master Mix (NEB), 1 µL of the barcode-attached forward primer (10 µM), 1 µL of the barcode-attached reverse primer (10 µM), and 28 µL of purified cDNA. The PCR program included 1 cycle of initiation at 95℃ for 10 min, followed by 30 cycles of denaturation at 95℃ for 15 s, annealing at 62℃ for 15 s, and extension at 65℃ for 7 min. The program ended with 1 cycle of final extension at 65℃ for 7 min. PCR amplicons were purified using the SPRI size selection method with a final elution volume of 30 µL. Note that size selection was not performed in the pilot run. The barcoded PCR amplicons were quantified using a Qubit fluorometer and pooled in equimolar quantities to a total of 1 µg DNA in a 48 µL sample (nuclease-free water was added if the total volume was less than 48 µL).

## Library preparation and MinION nanopore sequencing

To multiplex large numbers of samples, the dual barcode approach was adapted, and the library construction was prepared as previously described (34). The library preparation

involved the pooling of 12 amplicons, each possessing a unique barcode, using the native barcoding expansion (EXP-NBD104, NB01-NB07) and ligation sequencing kit (SEQ-LSK109). The full-length genome sequencing of enterovirus was performed on an ONT MinION sequencer. Two MinION (FLO-MIN106D) flowcells were used. Briefly, a total volume of 48 µL pooled DNA was added 3.5 µL buffer, 2 µL enzyme mix of NEBNext FFPE Repair Mix (M6630, NEB, Ipswich, MA, USA), and 3.5 µL buffer, 2 µL enzyme mix of NEBNext Ultra II End Repair/dA-Tailing Module (E7546, NEB, M6630, Ipswich, MA, USA). The mixture was incubated at 20°C for 30 min followed by 65°C for 30 min for DNA repairing and dA-tailing. The repaired and dA-tailed DNA was purified using an equal volume of Ampure XP (A63881, Beckman Coulter, Brea, CA, USA). Native barcodes were attached to the purified DNA, 22.5 µL, by mixing 25 µL NEB Blunt/TA Ligase Master Mix (M0367, NEB, Ipswich, MA, USA), 2.5 µL one of the Native barcodes and incubated at room temperature for 15 min, followed by an equal volume of Ampure XP cleanup steps. Different native barcode-labeled samples were pooled in equimolar amounts in a final volume of 65 µL (nuclease-free water was added if the total volume was less than 65 µL) and 5 µL sequencing adaptor, 20 µL of NEBNext Quick Ligation Module (E6056, NEB, Ipswich, MA, USA) buffer, 10 µL of Quick T4 ligase were added. The mixture was incubated at room temperature for 15 min. The DNA was purified using 40 µL Ampure XP and washed twice with 250 µL long fragment buffer for the final cleanup. 12 µL of DNA was mixed with 37.5 µL sequencing buffer and 25.5 µL loading beads, the pre-sequencing mix was loaded into a MinION SpotON flowcell R9.4.1 (FLO-MIN106) for sequencing. After the sequencing runs, Guppy (v5.0.11) was used to perform basecalling. The passed FASTQ files (minimum quality value of 9) were concatenated into a file named reads.fastq.

## nanoEV pipeline

The concatenated reads were filtered with SeqKit (v2.2.0) to retain reads with a length between 1,000 and 8,000 bp (35). The dual-barcode sequences generated by genera-tebcs.py (34) were used as a target to query the filtered reads using Minimap2 (v2.24) (36) by -k7 -A1 -m50 -w1 options. With a pairwise read mapping format (PAF) file output by Minimap2, the corresponding reads for a specific dual-barcode were generated by getbcfa.py. Besides, the barcoding sequences were trimmed out and the sequences containing 20-base poly(A) or poly (T) in the middle region (except the first 100 and the last 100 bases) were removed. After selecting the 200 longest sequences for each sample, the selected reads were self-aligned using Minimap2 to find overlaps longer than 6,000 bp between reads. With the self-aligned PAF file, a representative read was selected when it contained at least 40 (200 × 0.2) overlaps with other reads; another representative read was additionally selected if its overlapping reads had no intersection between the previous one. For each sample, consensus sequences were generated using Medaka (v1.4.3) along with the 200 sequencing reads and the representative sequen-ces. The consensus sequences output by Medaka were further polished using Homo-polish (v0.3.3) (37) along with enterovirus complete genome sequences. The polished sequences were searched against the enterovirus complete genome sequences (36) using BLASTN (2.13.0) (32) for enterovirus typing and were also translated for producing polyprotein and UP sequences. An iterative procedure was conducted for selecting a proper representative read whose alignment length was longer than 6,000 bp, and the lengths of the polyprotein and UP greater than 2,000 and 40, respectively, or having an identical polyprotein sequence between two successive iterations. Although three additional iterations were carried out at most to find out representative reads for a sample, the sequences for some samples were not successfully translated to polypro-teins. Two groups of reads for the samples were generated based on tetra-nucleotide frequencies of sequence (38), and each group of reads was analyzed as described previously. Finally, representative read ID, enterovirus type, sequence identity, alignment length, consensus sequence, and sequences of polyprotein and UP for each sample were

summarized in a file named Final_report.csv. The nanoEV pipeline is available at https://github.com/jade-nhri/nanoev.

## Phylogenetic analysis and recombination detection

All sequence alignments were performed using the MUSCLE method implemented in the MEGA program (39). Subsequently, the phylogenetic dendrograms were constructed using the neighbor-joining method. To detect recombination events, SimPlot (Version 3.5.1) was employed to generate similarity and bootscanning plots (30).

## ACKNOWLEDGMENTS

We thank Shu-Ting Luo, Wan-Yu Chung, and Yao Chang for administration and laboratory support.

The historical virus samples were originally collected in different studies. These studies were separately approved by the Institute Review Board (IRB) of Children's Hospital No. 1, Ho Chi Minh City, Vietnam (IORG0007285, FWA00009748), the Director of Health for Sarawak, Malaysia, and Taiwan CDC (IRB#110118). Informed consent was obtained from study participants' parents or guardians. The IRB of Taiwan CDC approved this study and waived informed consent based on Taiwan's Communicable Disease Control Act for collecting clinical specimens from patients with suspected notifiable communicable diseases.

This work was supported by grants from National Health Research Institutes (IV-111-GP-12 to F.-J. Chen and PH-111-PP05 to Y.-C. Liao, National Flagship Project 108-0324-01-19-11 to M.-S. Lee) and from Ministry of Health and Welfare, Taiwan (New South Bound Project MOHW110-OOIC-T-221-000001 to M.-S. Lee).

## AUTHOR AFFILIATIONS

[1]National Institute of Infectious Diseases and Vaccinology, National Health Research Institutes, Zhunan, Taiwan
[2]Department of Biological Science and Technology, National Yang Ming Chiao Tung University, Hsinchu, Taiwan
[3]Institute of Population Health Sciences, National Health Research Institutes, Zhunan, Taiwan
[4]Institute of Health and Community Medicine, Universiti Malaysia Sarawak, Sarawak, Malaysia
[5]Research and Diagnosis Center, Centers for Disease Control, Taipei, Taiwan

## AUTHOR ORCIDs

Chieh-Hua Lin  http://orcid.org/0000-0002-7920-2961
Min-Shi Lee  http://orcid.org/0000-0001-5914-3370
Yu-Chieh Liao  http://orcid.org/0000-0002-4360-7932

## FUNDING

| Funder | Grant(s) | Author(s) |
| --- | --- | --- |
| National Health Research Institutes (NHRI) | IV-111-GP-12 | Feng-Jui Chen |
| National Health Research Institutes (NHRI) | PH-111-PP05 | Yu-Chieh Liao |
| Ministry of Health and Welfare (MOHW) | 108-0324-01-19-11 | Min-Shi Lee |
| Ministry of Health and Welfare (MOHW) | MOHW110-OOIC-T-221-000001 | Min-Shi Lee |

## AUTHOR CONTRIBUTIONS

Yeh-Sheng Chien, Data curation, Investigation, Writing – original draft | Feng-Jui Chen, Conceptualization, Funding acquisition, Writing – original draft, Writing – review and editing | Han-Chieh Wu, Investigation, Methodology, Writing – original draft | Chieh-Hua Lin, Formal analysis, Investigation | Wen-Chiung Chang, Methodology, Resources | David Perera, Resources, Writing – review and editing | Jyh-Yuan Yang, Resources | Min-Shi Lee, Conceptualization, Funding acquisition, Resources, Writing – original draft, Writing – review and editing | Yu-Chieh Liao, Conceptualization, Data curation, Formal analysis, Funding acquisition, Investigation, Writing – original draft, Writing – review and editing

## DATA AVAILABILITY

The sequencing data generated from this study are publicly available in Figshare under https://doi.org/10.6084/m9.figshare.24129558.v1 and https://doi.org/10.6084/m9.figshare.24129729.v1.

## ADDITIONAL FILES

The following material is available online.

### Supplemental Material

**Supplemental tables (Spectrum02507-23-s0001.xlsx).** Tables S1 to S6.

### Open Peer Review

**PEER REVIEW HISTORY (review-history.pdf).** An accounting of the reviewer comments and feedback.

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
