## [Reviewer comments · Microbiology Spectrum]

Microbiology Spectrum

Cost-Effective Complete Genome Sequencing Using the MinION Platform for Identification of Recombinant Enteroviruses

Yeh-Sheng Chien, Feng-Jui Chen, Han-Chieh Wu, Chieh-Hua Lin, Wen-Chiung Chang, David Perera, Jyh-Yuan Yang, Min-Shi Lee, and Yu-Chieh Liao

Corresponding Author(s): Yu-Chieh Liao, National Health Research Institutes

Review Timeline:

Submission Date:	June 14, 2023
Editorial Decision:	August 4, 2023
Revision Received:	August 11, 2023
Accepted:	August 30, 2023

Editor: Day-Yu Chao

Reviewer(s): Disclosure of reviewer identity is with reference to reviewer comments included in decision letter(s). The following individuals involved in review of your submission have agreed to reveal their identity: Michael Owusu (Reviewer #1)

Transaction Report:

DOI: <https://doi.org/10.1128/spectrum.02507-23>

August 4, 2023

Dr. Yu-Chieh Liao
National Health Research Institutes
Institute of Population Health Sciences
35 Keyen Rd.
Miaoli County, Taiwan 350
Taiwan

Re: Spectrum02507-23 (Cost-Effective Complete Genome Sequencing Using the MinION Platform for Identification of Recombinant Enteroviruses)

Dear Dr. Yu-Chieh Liao:

Link Not Available

Sincerely,

Day-Yu Chao

Journals Department
Reviewer comments:

Reviewer #1 (Comments for the Author):

Authors have Cost-Effective Complete Genome Sequencing Using the MinION Platform for the Identification of Recombinant Enteroviruses. The manuscript appears well-written. I have few comments:

1. CODEHOP should be first defined in the abstract before writing this as an abbreviation.
2. Authors appear to only use nucleotide percentage identity as a means of assessing the performance of MinIONs compared to Illumina. Authors should consider using depth and quality of reads as additional variables to enable a good understanding of the utility of minION.

3. It was not clear if the authors obtained ethics clearance for this research. Authors should indicate this is available.
4. Line 400: Authors should rephrase the statement on the "bp \leq length \leq 8000 bp". This is not quite clear

Reviewer #2 (Comments for the Author):

In this manuscript entitled "Cost-Effective Complete Genome Sequencing Using the MinION Platform for Identification of Recombinant Enteroviruses", the authors used Oxford Nanopore MinION sequencer to generate the genome sequence of enterovirus. With this sequencing method, they identified co-infections and produced polyprotein sequence. They identified recombinant strains of EVA71 and CVA2, and performed phylogenetic analysis to investigate the evolution and spread of enterovirus subgenogroups. The authors sequenced 36 enterovirus isolates in a pilot run and 70 enterovirus isolates in a high accurate run. In general, this manuscript is well written.

Major point:

1. In the pilot run and high-accurate run, the authors used MinION to generate complete enterovirus sequences. More information and analyses of the sequencing data are required. The number of sequencing reads for each run, the percentage of enterovirus reads in the total reads, genome coverage for enterovirus, and sequencing depth etc.

2. The enterovirus sequence from MinION sequencer is 99.98% identical to that from the Illumina sequencer. This result is obtained from both the pilot and high-accurate runs. I'm a bit concern with the difference between MinION and Illumina enterovirus sequences. What are those differences? What's the source of these differences? Possible to reduce the sequencing errors?

3. The title of this manuscript is "Cost-Effective Complete Genome Sequencing Using the MinION Platform for Identification of Recombinant Enteroviruses". The current manuscript did not provide much information of the sequencing cost. I would suggest the author present detailed information of the cost, compare with illumine sequencing, and add some discussion.

Major point:

1. "next-generation sequencing" was one of the keywords. Nanopore sequencing?

2. Line 413, v0.3.3.

Staff Comments:

Preparing Revision Guidelines

Please return the manuscript within 60 days; if you cannot complete the modification within this time period, please contact me. If you do not wish to modify the manuscript and prefer to submit it to another journal, please notify me of your decision immediately so that the manuscript may be formally withdrawn from consideration by Microbiology Spectrum.

Responses to reviewers' comments:

Reviewer #1 (Comments for the Author):

Authors have Cost-Effective Complete Genome Sequencing Using the MinION Platform for the Identification of Recombinant Enteroviruses. The manuscript appears well-written. I have few comments:

1. CODEHOP should be first defined in the abstract before writing this as an abbreviation.

[Response] We appreciate the reviewer's suggestion; we have made the appropriate corrections accordingly. [line25]

2. Authors appear to only use nucleotide percentage identity as a means of assessing the performance of MinIONs compared to Illumina. Authors should consider using depth and quality of reads as additional variables to enable a good understanding of the utility of minION.

[Response] We are grateful for the reviewer's valuable suggestion. In conjunction with the inclusion of consensus sequences and their corresponding identities with the best-hit genomes in the "final_report.csv" file, the nanoEV pipeline has also generated read alignment files for individual samples. To assess the quality of these consensus sequences, these alignment files can be effectively visualized using tools such as IGV (<https://software.broadinstitute.org/software/igv/>) or Tablet (<https://ics.hutton.ac.uk/tablet/>). As indicated in Table 1, there are eight sequences displaying nucleotide percentage identities (to the best-hit genomes, sequenced by Illumina) lower than 99.98%. Hence, we would like to ascertain the accuracy of the consensus call. Following a thorough examination, all consensus sequences generated by Medaka were found to be accurate. Two instances of discordance (between Illumina and MinION) can be observed in the nanoEV Manual.

The discordance between Illumina and MinION

Seq of P01 vs. MF422535

```
Query  974  ATTTCACTCAAGACCCTGGCAAGTTCACACAACCTGTATTGGATGCTTTACGTGAAGCTG  1033
      |||
Sbjct  846  ATTTCACTCAAGACCCTGGCAAGTTCACACAACCTGTATTGGATGCATTACGTGAAGCTG  905

Query  2834  CGAGCGCTACAGGGTTTACCAAATGGGATATAGATATAATGGGGTATGCGCAATTGCGCA  2893
      |||
Sbjct  2706  CGAGCGCTACAGGGTTTACCAAATGGGATATAGATATAATGGGGTATGCGCAATTGCGCA  2765
```

Seq of P03 vs. MF422543:

```
Query 1292 CGATCACAGTTGCACCTTTGTGTTTCAGAGTTTGGCTGGATTACGACAAGCCGTAAAGGCAGG 1351
Sbjct 1616 CGATCACAGTTGCACCTTTGTGTTTCAGAGTTTGGCTGGATTACGACAAGCCGTAAAGGCAGG 1675
Query 2552 GCAAGCCAGATGGTAGGGAGGCTTTCCAGTGGCAATCCTCAACCAACCCATCTGTGTTTA 2611
Sbjct 2876 CCAAGCCAGATGGTAGGGAGGCTTTCCAGTGGCAATCCTCAACCAACCCATCTGTGTTTA 2935
```

3. It was not clear if the authors obtained ethics clearance for this research. Authors should indicate this is available.

[Response] The clearance has been described in the ACKNOWLEDGMENTS section. [line425-432]

4. Line 400: Authors should rephrase the statement on the "bp≤length≤8000 bp".

This is not quite clear

[Response] The statement has been corrected to: to retain reads with a length between 1000 bp and 8000 bp. [line 389]

3. The title of this manuscript is "Cost-Effective Complete Genome Sequencing Using the MinION Platform for Identification of Recombinant Enteroviruses". The current manuscript did not provide much information of the sequencing cost. I would suggest the author present detailed information of the cost, compare with illumine sequencing, and add some discussion.

[Response] We appreciate the suggestion provided by the reviewer. We have included **Supplement Table S6** to outline the consumption expense for nanopore sequencing. Moreover, in accordance with our previous publication (<https://jbiomedsci.biomedcentral.com/articles/10.1186/s12929-019-0541-x>), the outsourcing cost for Illumina sequencing amounted to USD500 per sample. **[line 253]**

Table S6 consumption expense for nanopore sequencing				
Reaction	Reagent	Unit price (USD)	Quantity	Subtotal (USD)
Reverse Transcription	SuperScript IV Reverse Transcriptase		5	70
PCR	LongAmp Taq 2X Master Mix		2	70
Cleanup	Agencourt AMPure XP - PCR Purification		1	70
Gap repair	NEBNext® FFPE DNA Repair Mix		11	7
End-repair/dA Tailing	NEBNext® Ultra™ II End-Repair / dA-tailing Module		13	7
Cleanup	Agencourt AMPure XP - PCR Purification		1	7
Barcode Ligation	NEB Blunt/TA Ligase Master Mix		14	7
Cleanup	Agencourt AMPure XP - PCR Purification		1	7
Adapter Ligation	NEBNext® Quick Ligation Module		38	1
Cleanup	Agencourt AMPure XP - PCR Purification		1	1
Library construction	ONT kits		148	1
Sequencing	ONT flowcell		900	1
Total Cost in USD				1927
AVG= 1927/70=27.5 (USD)				

Major point:

1. "next-generation sequencing" was one of the keywords. Nanopore sequencing?
2. Line 413, v0.3.3.

[Response] We have addressed these points and made the necessary corrections.

[line58, line404]

August 30, 2023

Dr. Yu-Chieh Liao
National Health Research Institutes
Institute of Population Health Sciences
35 Keyen Rd.
Miaoli County, Taiwan 350
Taiwan

Re: Spectrum02507-23R1 (Cost-Effective Complete Genome Sequencing Using the MinION Platform for Identification of Recombinant Enteroviruses)

Dear Dr. Yu-Chieh Liao:

Your manuscript has been accepted, and I am forwarding it to the ASM Journals Department for publication. You will be notified when your proofs are ready to be viewed.

Sincerely,

Day-Yu Chao
Editor, Microbiology Spectrum
